# Rethinking Sim2Real: *Lower* Fidelity Simulation Leads to *Higher* Sim2Real Transfer in Navigation

**Joanne Truong[1], Max Rudolph[1], Naoki Yokoyama[1],**
**Sonia Chernova[1], Dhruv Batra[1,2], Akshara Rai[2]**
[1]Georgia Institute of Technology, [2]Meta AI
{truong.j, maxrudolph, nyokoyama, chernova, dbatra}@gatech.edu
{akshararai}@fb.com

**Abstract:** If we want to train robots in simulation before deploying them in reality, it seems natural and almost self-evident to presume that reducing the sim2real gap involves creating simulators of increasing fidelity (since reality is what it is). We challenge this assumption and present a contrary hypothesis – sim2real transfer of robots may be improved with *lower* (not higher) fidelity simulation. We conduct a systematic large-scale evaluation of this hypothesis on the problem of visual navigation – in the real world, and on 2 different simulators (Habitat and iGibson) using 3 different robots (A1, AlienGo, Spot). Our results show that, contrary to expectation, adding fidelity does not help with learning; performance is poor due to slow simulation speed (preventing large-scale learning) and overfitting to inaccuracies in simulation physics. Instead, building simple models of the robot motion using real-world data can improve learning and generalization.

**Keywords:** Sim2Real, Deep Reinforcement Learning, Visual-Based Navigation

## 1 Introduction

The sim2real paradigm consists of training robots in simulation (potentially for billions of simulation steps corresponding to decades of experience [1]) before deploying them in reality. The last few years have seen significant investments – the development of new simulators [2–12], curation and annotation of 3D scans and assets [13–15], and development of techniques for overcoming the sim2real gap [16–19] – resulting in a number of successful demonstrations of sim2real transfer [20–25]. However, no simulator is a perfect replica of reality and the main challenge in this paradigm is overcoming the sim2real gap, defined as the drop in a robot's performance in the real-world (compared to simulation). It seems natural and almost self-evident to presume that reducing this sim2real gap involves creating simulators of increasing physics fidelity, and this sometimes forms the default operating hypothesis of the field.

We challenge this convention and present a counter-intuitive idea – sim2real transfer of robots may be improved not by increasing but by *decreasing* simulation fidelity. Specifically, we propose that instead of training robots entirely in simulation, we use classical ideas from hierarchical robot control [26] to decompose the policy into a 'high-level policy' (that is trained solely in simulation) and a 'low-level controller' (that is designed entirely on hardware and may even be a black-box controller shipped by a manufacturer). This decomposition means that the simulator does not need to model low-level dynamics, which can save both simulation time (since there is no need to simulate expensive low-level controllers), and developer time spent building and designing these controllers.

We conduct a systematic large-scale evaluation of our hypothesis on the task of PointGoal (visual) Navigation [27] in unknown environments – using 2 simulators (Habitat and iGibson) and 3 different robots (A1, AlienGo, Spot). We train policies using two physics fidelities – kinematic and dynamic. Kinematic simulation uses abstracted physics and 'teleports' the robot to the next state using Euler integration; kinematic policies command robot center-of-mass (CoM) linear and angular velocities. Dynamic simulation consists of rigid-body mechanics and simulates contact dynamics (via Bullet [12]); dynamic policies command CoM linear and angular velocities, which are converted to robot joint-torques by a low-level controller operating at 240 Hz. We find that across all robots, a kinematically trained policy outperforms dynamic policies, *even when evaluated using dynamic simulation*

6th Conference on Robot Learning (CoRL 2022), Auckland, New Zealand.

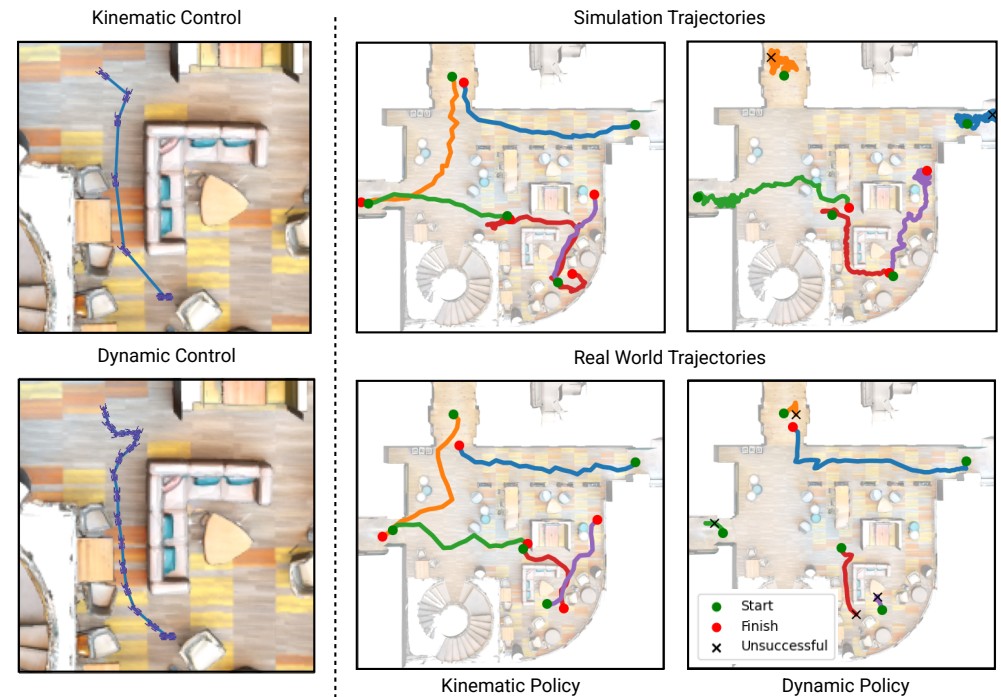

Figure 1: **Left:** We train visual navigation policies at two levels of fidelity – kinematic and dynamic. In kinematic control (top), the robot is 'teleported' to the next state using Euler integration. In dynamic control (bottom), the robot's velocity commands are converted to leg joint-torques and rigid-body physics is simulated at 240Hz. **Right:** We evaluate the kinematic and dynamic trained policies in simulation (top) and the real-world (bottom) across 5 identical episodes. The kinematic policy achieves a 100% success rate in all 5 episodes, and the robot takes similar paths in both simulation and the real-world. On the other hand, the dynamic policy achieves a 20-60% success rate, and the trajectories taken in simulation and the real-world do not correlate, pointing towards a larger sim2real gap.

*and control.* Additionally, we show that the trained kinematic policy can be transferred to a real Spot robot, which ships with manufacturer-provided 'black-box' low-level controllers that cannot be accurately simulated. In contrast, dynamic policies fail to achieve efficient navigation behavior on Spot, due to the sim2real gap and less simulation experience.

The reasons for these improvements are perhaps unsurprising in hindsight – learning-based methods overfit to simulators, and present-day physics simulators have approximations and imperfections that do not transfer to the real-world. A second equally significant mechanism is also in play – lower fidelity simulation is typically faster, enabling policies to be trained with more experience under a fixed wall-clock budget. Even when the kinematic policies were trained for 2.3× less wall-clock time than the dynamic policies (with the same compute), the kinematic policies were able to learn from 10× the amount of data. While our results are presented on legged locomotion and visual navigation, the underlying principle – of architecting hierarchical policies and only training the high-level policy in an abstracted simulation – is broadly applicable. We hope that our work leads to a rethink in how the research community pursues sim2real and in how we develop the simulators of tomorrow. Specifically, our findings suggest that instead of investing in higher-fidelity physics, the field should prioritize simulation speed for tasks that can be represented with abstract action spaces.

## 2   Related Work

**Visual Navigation.** Recent works have shown that large-scale indoor environments and simulators like Habitat [2, 3] and iGibson [4] can enable end-to-end learning of navigation policies from large amounts of agent- or expert-generated data [28–30] on simple, wheeled systems. This is in contrast to the typical mapping and planning paradigm used in classical robotics, which can suffer when the quality of maps is low [31] or requires expensive equipment like LiDAR [32]. In this work, we show that such end-to-end learning is also possible for complex, legged robots.

**Sim2real for Legged Robots.** Sim2real quadrupedal locomotion has been widely studied in the past several decades [22, 33–36], with most learning low-level skills in simulation and transferring them to hardware [37], or adapting them online to reduce the sim2real gap [38, 39]. However, these policies are typically blind, and use only proprioceptive sensors on the robot to determine actions [23, 25]. In contrast, an autonomous robot needs to respond to its environment, and take visual input into account. Some works have proposed learning visual policies in simulation and applying them to the real-world [22, 24, 40], and other works leverage expensive LiDAR sensors for external sensing [41]. These works use learned or hand-designed physically simulated low-level controllers; we show that physics simulations can be detrimental to learning high-performing sim2real policies, even for complex legged robots. Work from [42] also utilize similar simulator simplifications to increase simulation speed in training navigation policies for 1 robot in a single room, but does not discuss how this formulation affects sim2real transfer. In our work we present a rigorous (multi-robot, multi-simulator) study of the effect of different simulation fidelities on visual navigation.

**Abstracted Task-space Learning.** Abstracted (hierarchical, high-level) action spaces are common in robotics literature. Examples include task and motion planning for manipulation [43–45], legged locomotion [46, 47], navigation [20], etc. Several works reason over symbolic actions like pick and place, or hierarchical policies with discrete/continuous attributes [33–35, 48, 49], or even abstracted dynamics models [36]. While the ideas of abstracted/hierarchical policies are fairly common, typically both the high- and low-level policies are learned in simulation and transferred to reality [33, 34, 36], often augmented with techniques like domain randomization [37] and real-word adaption [18]. Instead, we use an abstracted simulator, which does not model low-level physics, and learn high-level policies that are transferred to the real-world in a zero-shot manner.

## 3 Experimental Setup

**Task: PointGoal Navigation.** In the task of PointGoal Navigation [27], a robot is initialized in an unknown environment and is tasked with navigating to a goal coordinate without access to a pre-built map of the environment. The goal is specified relative to the robot's starting location for the episode (i.e., "go to $\Delta$x, $\Delta$y"). The robot has access to an egocentric depth sensor and an egomotion sensor (sometimes referred to as GPS+Compass in this literature) from which the robot derives the goal location relative to its current pose. An episode is considered successful when the robot reaches the goal position within a success radius (typically half of the robot's body length). The robot operates within constraints of maximum number of steps per episode (150 for Spot) and velocity limits ($\pm$ 0.5 m/s for linear and $\pm$ 0.3 rad/s for angular velocities on Spot). We linearly scale the linear and angular velocity limits for A1 and Aliengo to be proportional to the length of each robot's leg, and inversely scale the maximum number of steps allowed. In effect, smaller robots have a smaller maximum allowed velocity to improve stability during execution, but are allowed more steps to reach the goal. The exact parameters used for each robot is described in the appendix. For evaluation, we report the success rate (SR), and Success inversely weighted by Path Length (SPL) [27], which measures the efficiency of the trajectory taken with respect to the ground-truth shortest path.

**Robot Platforms.** We study visual navigation for 3 quadrupedal robots – A1 and Aliengo from Unitree [50], and Spot from Boston Dynamics (BD) [51] in simulation. In the real-world, we show sim2real transfer of the learned navigation policies to Spot. To have a consistent camera setup across all the robots, we attach an Intel RealSense D435 camera to Spot in the real-world, and use this camera for visual inputs to the policy. In our hardware experiments, we want to measure how often our sim2real policies lead to collisions without jeopardizing safety. We achieve this balance as follows: the BD collision-avoidance capability is kept turned on, set to trigger at a tight threshold of 0.10m. Next, we track the number of times the robot comes within 0.20m of any obstacle (as measured by any of the 5 onboard depth cameras). This gap (between 0.20m and 0.10m) allows us to record possible collisions while preventing actual ones. While the BD API allows for high-level navigation without access to a map, it cannot navigate around obstacles autonomously, without a map. In our work, we consider complex, long-range navigation paths (up to 30m) in cluttered environments with many obstacles; the goals are unreachable with the just BD navigation API.

**Simulation Environments.** We use two simulation platforms – Habitat [2, 3] and iGibson [4] for training and evaluation. Both simulators support rendering of photorealistic environments; Habitat uses a low-level (C++) integration with the Bullet physics engine [12], while iGibson leverages Py-Bullet, the Python-based integration of Bullet. Thus, while the underlying physics engines between

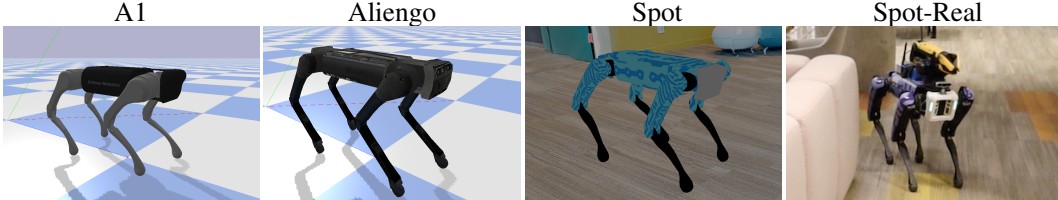

| A1 | Aliengo | Spot | Spot-Real |

Figure 2: Robots used for training and evaluation.

the two are the same, Habitat runs $\sim 1200\%$ faster than iGibson [3]. This allows us to train policies faster with Habitat than with iGibson even when using identical policies and compute.

**Dataset.** For training and evaluation, we use a combination of the Habitat-Matterport (HM3D) [13] and Gibson [52] 3D datasets. The two datasets combined consist of over 1000 high-resolution 3D scans of real-world indoors environments, and consists of realistic clutter. We generate training and evaluation episodes compatible with our robots for the HM3D and Gibson scenes following the procedure described in [2]. Specifically, we restrict the geodesic distance from the start and positions to be between 1 and 30m, and increase navigation complexity by rejecting paths that consist of near-straight lines, with few obstacles. As described in [2], both of these heuristics result in complex, but navigable paths. Additionally, we check for collisions along the sampled paths using the URDF of the largest robot (Spot) to ensure that all paths are navigable.

**Real-World Test Environment.** The real-world evaluation environment, LAB, is a $325m^2$ lobby in a commercial office building. The lobby contains furniture such a couches, cushions, bookshelves and tables. We specify a set of 5 waypoints as the start and end locations for the navigation episodes in LAB with an average episode length of 10m. We match the furniture layout to the position captured in the 3D scan (Figure 3) to run identical evaluation experiments in both simulation and the real-world. The scan of LAB is not part of training.

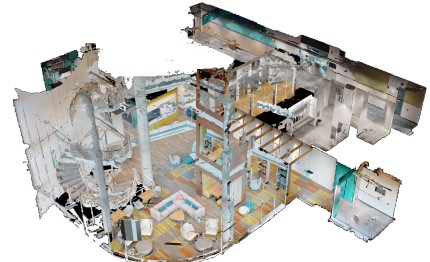

Figure 3: The real-world testing environment is a part of a large commercial building and contains clutter from furniture such as tables, bookshelves, and couches.

## 4   Kinematic and Dynamic Control for Visual Navigation

As illustrated in Figure 4, our proposed approach is hierarchical, with (1) a high-level visual navigation policy that commands desired center of mass (CoM) motion at 1Hz, and (2) a low-level controller that follows this desired motion. We consider controllers at two levels of abstraction – 'kinematic' and 'dynamic'. The kinematic controller simply integrates the desired velocity and outputs a CoM position at 1Hz; kinematic simulation then teleports the robot to the desired state. The dynamic controller uses a low-level controller that commands joint torques at 240Hz; dynamic simulation models rigid-body and contact dynamics via Bullet (with a physics step-size of 1/240 sec). We provide details of all three of these pieces (high-level policy, kinematic and dynamic controllers) next.

**High-level Visual Navigation Policies.** The high-level policy takes as input an egocentric depth image, and the goal location relative to the robot's current pose. The output of the policy is a 3-dimensional vector, representing the desired CoM forward, lateral, and angular velocities $(V_x, V_y, \omega)$. The neural network architecture consists of a ResNet-18 visual encoder and a 2-layer LSTM policy. Using a recurrent policy allows the policy to learn temporal dependencies through the hidden state. The final layer of the policy parameterizes a Gaussian action distribution from which the action is sampled. The policy is trained using DD-PPO [1], a distributed reinforcement learning method, in both the Habitat and iGibson simulators. Our reward function is derived from [22], with an added penalty for backward velocities, which can lead to collisions and hurts performance.

**Kinematic Control and Simulation.** In kinematic control, the final state of the robot is calculated by integrating the desired CoM velocity commanded by the high-level navigation policy at 1Hz. The robot is directly moved to the desired pose, without running a physics simulation. In both Habitat and iGibson, the robot is kept in place if being at the new desired state would result in a collision.

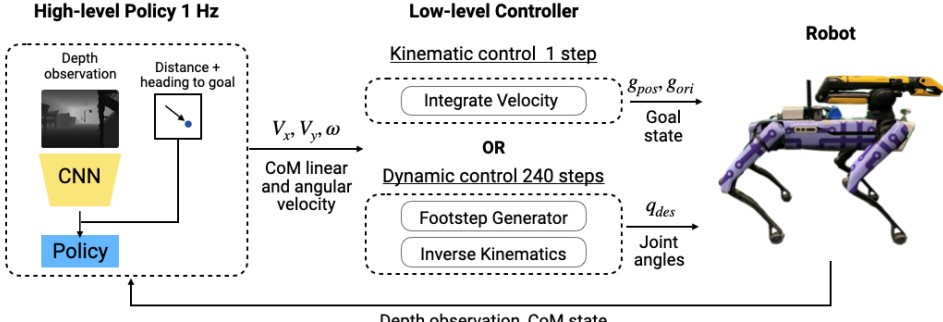

Figure 4: Our architecture for PointGoal Navigation on a legged robot. A high-level visual navigation policy predicts CoM linear and angular velocities. The velocities are passed into either a kinematic or dynamic low-level controller to step the robot in simulation. In the real-world, we directly send the velocity commands from the high-level policy to the robot, and uses the low-level controller from Boston Dynamics for movement.

The objective of the kinematic control is to abstract away the low-level physics interactions between the robot and its environment. This has two advantages: (1) it avoids the need to accurately model low-level controllers, especially for closed-source robots like Spot; (2) it enables faster simulation speed by avoiding high-frequency physics integration, conducive to model-free RL that requires large amounts of experience. On the other hand, teleporting the robot to the desired state might remove necessary dynamics, such as poor tracking of low-level controllers. In Section 5, we propose how to incorporate such low-level characteristics into a kinematic simulation using real-world data.

**Dynamic Control in Simulation and Hardware.** We experiment with two different low-level dynamic controllers for quadruped robots. The first is an expert-designed Raibert-style controller from [22], which consists of a footstep generator and an inverse kinematic solver that commands desired joint angles from CoM velocities. The joint angles are converted to joint torques using a linear feedback controller, and applied to the simulation. This controller was shown to achieve sim2real transfer for A1 [22]. However, on other robots in our experiments, it shows relatively poor tracking of high-level commands. Thus, we also experiment with another model-predictive control (MPC) dynamic controller from [38], which commands joint torques directly. This controller has been applied to real-world A1 robot [53, 54] and shows better tracking of desired velocities for our test robots, as compared to the Raibert controller from [22]. However, MPC is prohibitively slow and cannot be used for training RL policies. Thus, we use Raibert for training dynamic policies, but evaluate using MPC. [1] This difference in train and evaluation dynamics controllers has multiple purposes: (1) the evaluation using MPC improves performance of most policies, including dynamic policies, due to its better ability to track high-level commands; (2) the difference between the two dynamic controllers in simulation is also a proxy for the difference between our low-level controllers and closed-source controllers from Spot. If a dynamic policy cannot transfer from Raibert to MPC, it has a low chance of transfer to Spot which has black box BD controllers, or even other robots in the real-world.

Both dynamic controllers model the low-level physics interactions between the robot and the environment. This makes them considerably slower than the kinematic controller, making training RL policies challenging. Moreover, for Spot, the low-level controller implementation is not openly available, making it hard to reproduce the low-level controller in simulation. For commercial legged robots that come with black-box controllers, kinematic simulations are the ideal fidelity for learning navigation policies. Our experiments in Section 5 show that the added fidelity of dynamic controllers does not benefit policy learning, or sim2real transfer.

**Impact of Low-level Controllers on Policy Learning.** The low-level controller used in dynamic simulations can have a significant impact on the learned policy. Low-level controllers both in sim and real are biased, and the status quo is to make them biased in the same way, as the policy learns to compensate for the bias error. For example, [55] add learned actuation noise to their simulation, while [37] measure hardware characteristics and add them to the simulation. However, given the high-dimensional nature of low-level physics, it is very difficult to ensure that the biases incorporated in simulation actually hold for a larger range of motions that the data was collected on. Thus, there are several iterations of data-collection, bias updates, training and deployment needed for good

---

[1]Evaluation using Raibert [22] can be found in the appendix.

performance. Instead, kinematic controllers are unbiased by design, and can easily incorporate hardware bias through low-dimensional CoM motion noise models created from a small amount of real-world data, as shown in our experiments in Section 5.

# 5 Results and Analysis

In this section, we first study generalization of visual navigation policies across simulators (trained in one sim, tested in another) and across controllers (trained with one controller, tested with another). This shows the importance of fast simulation for learning high-level policies by comparing performance of kinematic and dynamic policies trained for the same wall-clock time. Next, we examine the performance of the different policies at zero-shot sim2real transfer on the Spot robot.

**How large is the sim2sim gap? High for dynamic, and low for kinematic policies.** We exhaustively study the combinatorial space of experiments – policies trained under 2 training conditions (with kinematic and dynamic simulation) × 2 evaluation conditions (kinematic and dynamic simulation) × 2 simulators (Habitat and iGibson) × 3 robots (A1, Aliengo, Spot). For each condition, we train and report results with 3 random seeds. Each policy is trained using 8 GPUs for 3 days, resulting in a cumulative training budget of 6,912 GPU-hours (288 GPU-days). The average success rates are presented in Figure 5. Rows represent the evaluation conditions as tuples (simulator, fidelity), while columns represent the training conditions. We evaluate all policies across 1,100 episodes from 110 unique scenes in the HM3D + Gibson validation split.

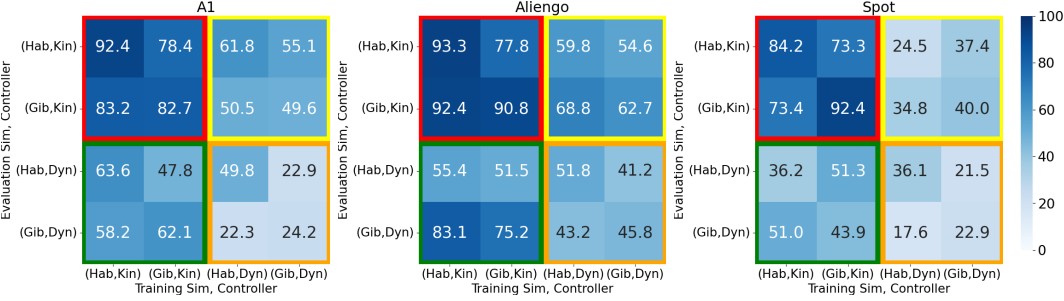

Figure 5: Average success rates for sim2sim and kinematic2dynamic transfer for A1, Aliengo and Spot. We see that the kinematic trained policies perform the best overall (red quadrants), and also often outperform the dynamic trained policies, even when evaluated using dynamic control (green quadrants vs. orange quadrants).

We make two key observations here:

1. **Kinematic-trained policies perform best overall, for all robots.** In all cases, kinematic policies outperform the dynamic policies, *even when evaluated using dynamic control*, *e.g.* 62.1% SR for A1 in (iGibson, Kinematic) *vs.* 24.2% SR in (iGibson, Dynamic), Fig. 5, left. This is a surprising result because the kinematic policies are being evaluated in an out-of-distribution setting, which was never seen or accounted for during training. On the other hand, the dynamic policies are being evaluated in the domain that they were trained in, hence do not require control-related generalization.

2. **Dynamic policies are not robust to different dynamic simulations.** The dynamic policies from the two simulations observe significant performance drops when evaluated in the other dynamic simulation. This points to the dynamic policies overfitting to the simulator dynamics during training, failing to generalize to a new setting, see *e.g.* column 3, rows 3 and 4; 49.8% SR for A1 in (Habitat, Dynamic) vs. 22.3% SR in (iGibson, Dynamic). (iGibson, Dynamic) shows poor performance in both iGibson and Habitat, with slightly poorer performance in Habitat. This sensitivity to simulation makes training dynamics policies difficult, especially when the controller for the real-world robot is unknown. Even if the real-world controller is known, simulation physics and real-world are different, and sim2real transfer of the learned policy can suffer (as evidenced by low sim2sim transfer). On the other hand, kinematic policies, that have been trained with no physics, can generalize to the different dynamic controllers. Both of these results go to show that not only kinematic trained policies are able to learn the task well, they have learned to reason without overfitting to simulation physics, making their chances of successful sim2real transfer high.

**Why do kinematic-trained policies outperform dynamic ones? Scale.** We plot the evaluation performance of both policies in Habitat kinematic and dynamic simulation in Figure 6. We train

both policies to convergence– 3 days for the kinematic policies, and 7 days for the dynamic policies. While the kinematic policies are trained for 2.3× less wall-clock time, they still outperform the longer trained dynamic policies, even when evaluated out-of-distribution using dynamic control (+12% SR). Kinematic training is much faster than training dynamically (right, Fig. 6); with kinematic training, the robot is able to learn from approximately 10× more steps of experience (500M steps vs. 50M steps). This increased experience allows the kinematic policies to learn intelligent high-level reasoning. We contend that for any computational budget, there will always be more complex tasks that are bottlenecked by that budget. Wall-clock time is the true limitation for learning-based sim2real approaches (not experience, as different simulators have different speeds).

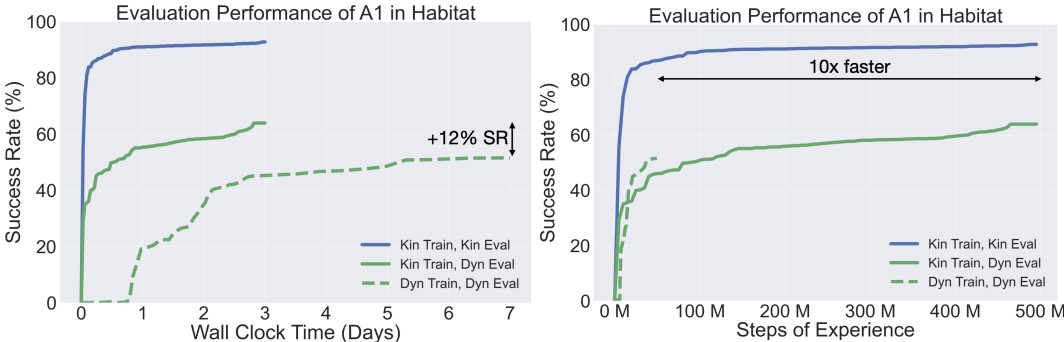

Figure 6: Success rate of PointNav policies with A1 trained and evaluated in Habitat with kinematic or dynamic control. Left: Kinematic policies outperform the dynamic trained policies (+12% SR), even when evaluated using dynamic control. Right: Using kinematic control, we can train our robot for 10× more steps of experience than with dynamic control under identical compute budgets, despite training for 2.3× less wall-clock time.

**How large is the sim2real gap for kinematic and dynamic trained policies?** We evaluate the kinematic and dynamic policies on a Spot robot in the novel LAB environment described in Section 3. Note that scans of LAB were not part of training. We evaluate 3 seeds of each policy over 5 episodes in the real-world and report the average success rate (SR) and Success weighted by Path Length (SPL) [27] in Table 1 (reported as a percentage for readability). Each control type is tested in 15 real-world episodes; one run of the Spot robot navigating LAB is shown in Figure 7. Success in the real-world is measured by computing final distance from the goal position using egomotion estimates provided by the Boston Dynamics SDK.

| Train | | | Simulation | | Reality | | | | Sim2Real Gap | |
|---|---|---|---|---|---|---|---|---|---|---|
| Simulator | Control | Noise | SR | SPL | SR | SPL | # Act. | # Coll. | SR | SPL |
| Habitat | Dynamic | - | 60.0 | 38.7 | 40.0 | 28.2 | 107.9 | 41.2 | 20.0 | 10.5 |
| iGibson | Dynamic | - | 20.0 | 14.0 | 67.7 | 46.6 | 76.8 | 12.9 | -47.7 | 32.6 |
| Habitat | Kinematic | - | 93.3 | 76.9 | **100.0** | 82.7 | 26.4 | 3.1 | -6.7 | -5.8 |
| iGibson | Kinematic | - | **100.0** | **90.6** | **100.0** | 83.2 | 33.1 | 4.5 | 0.0 | 7.4 |
| Habitat | Kinematic | Decoupled | 80.0 | 72.0 | **100.0** | 87.8 | 27.1 | **2.5** | -20.0 | -15.8 |
| Habitat | Kinematic | Coupled | 80.0 | 74.1 | **100.0** | **88.8** | **22.7** | 2.8 | -20.0 | -14.7 |

Table 1: Zero-shot sim2real transfer performance for the visual navigation policies. Success rate (SR) and path efficiency (SPL) are high for kinematic policies, while dynamic policies have lower performance due to the dynamics gap between the low-level control in training and the controller on the robot in the real-world.

As reported in Table 1, all kinematic policies achieve a high success rate of 100% and SPL of 82-83% (rows 3 and 4). On the other hand, the success rate drops to 40-67% for the dynamic policies (rows 1 and 2). We notice that the dynamic policies typically commanded lower velocities, and often get stuck around obstacles (Figure 8). This is shown in the higher number of actions commanded and higher collision count for both dynamic policies; on average, a dynamic policy trained in Habitat took 107.9 actions, and collided 41.2 times (row 1, columns 8 and 9), whereas a kinematic policy also trained in Habitat took 26.4 actions, and collided 3.1 times (row 3, columns 8 and 9). We attribute this to the impoverished experience of the dynamic policies; the policies did not learn robust navigation policies that could avoid obstacles during navigation. Additionally, they overfit to the low-level behavior, which can be unstable at high velocities in sim, but not on hardware. Figure 8 (left) shows that the kinematic policy commands higher forward velocities, while the dynamic

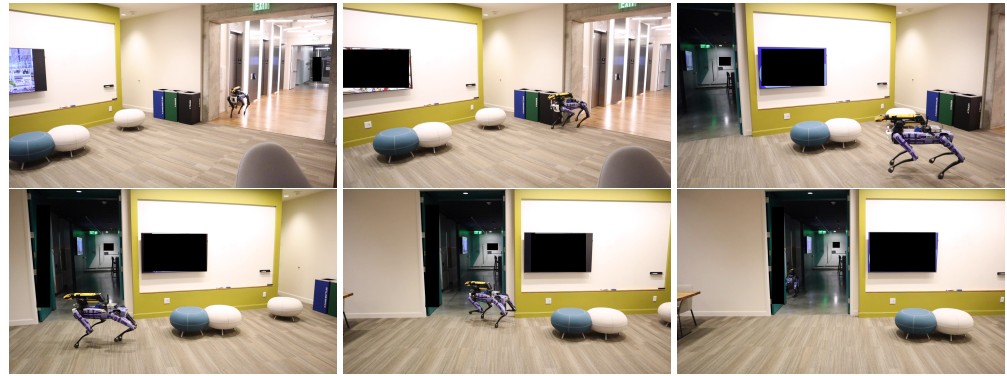

Figure 7: One run of the Spot robot navigating the real-world LAB environment using a kinematically trained policy from AI Habitat. The robot successfully navigates a hallway, moves around furniture and turns into the next hallway before stopping. In contrast, the native BD controllers without a map can only reach visible goals.

policy commands slower velocities (right), which are often not achieved by the robot likely due to an obstacle. [2] Successfully executed commands appear on the diagonal.

To improve kinematic simulation fidelity, we model actuation noise (difference between commanded and true velocity) on Spot and use it during kinematic training, similar to [16]. We collect 6,000 samples of decoupled (linear and angular velocities are actuated separately) and coupled (linear and angular velocities are actuated together) actuation noise. The parameters for noise in each dimension, and details about data collection and modeling can be found in the appendix. During training, we sample from the Gaussian distribution for each dimension, and add it to the policy's predicted velocity.

We see that policies trained with noise also achieve 100% success in the real-world (rows 5 and 6), and are able to increase path efficiency (SPL) (4.6% using decoupled (row 4 vs. 5), and 5.6% using coupled actuation noise (row 4 vs. 6)). The number of collisions and commanded actions are also lower for these policies, compared to kinematic policies trained with no noise (22.7 actions and 2.8 collisions vs. 26.4 actions and 3.1 collisions). These improvements are due to the added robustness that

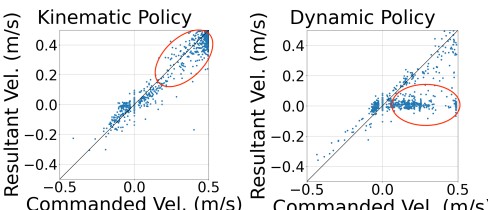

Figure 8: Commanded vs. resultant velocities during real-world trajectory rollouts for dynamically and kinematically trained policies.

training with noise provides – uncertainty during training forces the policy to take less risky actions resulting in fewer collisions in the real-world.

## 6 Conclusion, Limitations, and Future Work

In this work, we study the role of simulation fidelity for sim2real of visual navigation policies on three simulated and one real legged robot. Contrary to expectations, we find that higher simulation fidelity does not enable learning better high-level visual navigation policies. Dynamic policies tend to overfit to low-level simulation details, resulting in poor transfer to the real-world. On the other hand, kinematic policies are able to generalize well. These results raise important questions about the need for simulation fidelity for sim2real, especially in abstracted action spaces.

One limitation of this work is that we assume access to a robust 'black box' controller on hardware. While most robots come shipped with manufacturer-provided controllers, the level of accuracy may differ between robots, and more robust noise modeling may be needed to better characterize the actuation noise. In the future, we plan to improve the modeling of real-world actuation noise by using a neural network conditioned on previous states and actions of the robot. Another limitation of this work is that we specifically address only tasks that can be abstracted with high-level actions, like navigation or object rearrangement (pick, place, open, close). We agree that there are many tasks that cannot be learned in a kinematic simulation – e.g. tasks that require reasoning about low-level interactions with the environment, like dexterous manipulation, or low-level walking controllers. It is essential to create high-fidelity simulators for sim2real on such tasks.

---

[2]Actual velocity is measured using the Boston Dynamics SDK.

## Acknowledgments

The Georgia Tech effort was supported in part by NSF, ONR YIP, ARO PECASE. JT was supported by an Apple Scholars in AI/ML PhD Fellowship. The views and conclusions contained herein are those of the authors and should not be interpreted as necessarily representing the official policies or endorsements, either expressed or implied, of the U.S. Government, or any sponsor.

**License for dataset used** Gibson Database of Spaces. License at http://svl.stanford.edu/gibson2/assets/GDS_agreement.pdf

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
