# OpenReview forum: "Rethinking Sim2Real: Lower Fidelity Simulation Leads to Higher Sim2Real Transfer in Navigation"
_robot-learning.org/CoRL/2022/Conference — CoRL 2022 Poster_

### Official Review · Reviewer_BmX5 · 2022-07-27

**Originality:** Very Good
**Technical Quality:** Good
**Clarity Of Presentation:** Very Good
**Impact:** 3

**Recommendation:**

Weak Accept: I recommend accepting the paper, but will not argue for my recommendation if the majority of other reviewers have a different opinion.

**Summary:**

The paper challenges the common assumption that sim2real transfer benefits from increasing physics fidelity and presents a contrary hypothesis that sim2real transfer can be improved with lower physics fidelity (abstracted physics). The authors conduct a systematic evaluation with three legged robots on two simulation platforms in the domain of visual PointGoal navigation. The authors claim that higher fidelity hurts performance because 1) it’s slow and therefore prevents large-scale training, 2) leads to overfitting to inaccuracies in simulation physics.

**Issues:**

To help present a more balanced, impartial view of whether to increase/decrease simulator fidelity for researchers and practitioners, here are some concrete suggestions:

Training:
Model A. (Done already) Kinematic simulation: assume a perfect controller, fast to simulate
Model B. (Done already) Kinematic simulation with actuation noise modeled from real-world data: fast to simulate, seems like a very promising way to close the sim2real gap.
Model C. (New) Dynamic simulation with Raibert-style controller: slow to simulate, train until convergence*

Evaluation:
Eval A. (New) Dynamic simulation with the actual controller of the robot (if possible, it seems that the controller for A1 is open-sourced?)
Eval B. (Done already) Real-world with the actual controller of the robot

The research questions that hopefully these additional experiments can help answer are:
Model C: how long does it take for the policies trained in dynamic simulation to fully converge? If the researchers have more training time budget, is this a better option?
Model C: once converged, how is its performance compared to kinematic policies? Does the conclusion of the paper still hold? Does “overfitting to inaccurate physics” actually happen?
Eval A: this is important because it isolates the actuation sim2real gap (there is no visual sim2real gap). It will solidify the authors’ claim that higher physics fidelity leads to “overfitting” of the inaccuracy of physics simulation IF model A outperforms model C in Eval A. Also, if model B is the best in Eval A, the authors might claim that doing kinematic simulation while modeling actuation noise with real-world data is the best way to close the sim2real gap and be robust against different testing dynamics (environment or black-box controller), which would be a great lesson to share with the community. Right now, this argument is not bulletproof yet.
Eval B is obviously the final test, with some confounding factor of visual sim2real gap. But still, it would be great to show model B outperform model A and C here.
In the very end, it would be great to present a fair assessment of when to increase/decrease physics fidelity for better sim2real performance. I am not fully convinced that the answer is as black-and-white as presented in the current paper.

Two minor comments / questions
- What causes the difference in dynamic simulation between Habitat and iGibson? Aren’t they both using bullet as the underlying physics engine? Maybe it’s two different versions of bullet?
- It would be great if the authors can be consistent with the name of the simulators used. iGibson is mentioned in some places and Gibson is mentioned in others. Given the citation, it should probably always be iGibson?


**Quality Of The Limitations Section:**

Limitations are addressed clearly

**Reviewer Expertise:**

5: The reviewer is absolutely certain that the evaluation is correct and very familiar with the relevant literature

**Robotics Focus:**

Sufficient demonstration on hardware

**Strengths And Weaknesses:**

The paper conducted a large-scale study of sim2real transfer of legged robot locomotion: across 3 different robots, on 2 different simulators. All combination of (train, test) x (sim, fidelity) has been thoroughly evaluated. The authors also demonstrated the performance of their policy on real hardware in an unseen environment. More importantly, the paper proposes a very counterintuitive conclusion that challenges the almost self-evident assumption that sim2real gap involves creating simulators of increasing fidelity. This topic is very important and relevant for roboticists and practitioners interested in sim2real transfer for locomotion tasks and beyond. The lessons learned here have the potential to be widely applicable to the wider robotics domain when action abstraction can be applied (e.g. grasping, manipulation, etc). It’s appreciated that the authors take the effort to conduct this thorough study in sim and in real to answer this research question: do we need a higher-fidelity simulator?

However, the paper also has a major weakness that needs to be addressed. The authors claim that high fidelity simulation hurts performance because 1) it’s slow and therefore prevents large-scale training, 2) leads to overfitting to inaccuracies in simulation physics. Unfortunately, these two aspects are entangled together, and therefore the conclusion is not entirely convincing. To be more specific, in Figure 6 (right plot), it’s very clear that the policies trained with dynamic simulation (dotted lines) have not converged at all. The authors gave a training budget of 3 days and showcased that the policies trained with kinematic simulation (converged) outperform those trained with dynamic simulation (not converged). Their argument is somewhat one-sided, because what if the practitioners have a training budget of 14 days, rather than 3 days? In this case, maybe training with dynamic simulation is better?

Furthermore, the authors argue that policies trained with dynamic simulation “overfit” to the simulator dynamics, failing to generalize to a new setting (e.g. a black-box controller of a real robot and the real-world dynamics of the environment). The validity of this argument is questionable, because, again, this is based on the results of evaluating policies that have not converged yet. It’s very unclear if the policies “overfit” to the specific simulator dynamics, or if the policies actually underfit: they haven’t even had enough training examples to fit the training simulator, not to mention the other test simulator. In general, we should be very careful about any conclusion drawn from half-trained policies because the evaluation results from non-converged neural network-based policies can be arbitrary and meaningless.

It would be great if the authors can disentangle their two arguments and present a more impartial view for the practitioners and researchers to help them decide whether to increase or decrease the fidelity of their simulator. Feel free to check out the “Issues” section below for some concrete suggestions.



**Summary Of Recommendation:**

As mentioned above, although the problem domain is extremely relevant and interesting, and the authors did an amazing job of running thorough experiments across robot platforms and simulators and presenting their findings in the paper, the main conclusion of the paper is a bit one-sided and begs for further investigation. Since this paper is very focused and has essentially one strong message to deliver, this message has to be bullet-proof. If the authors can address the issues mentioned below, I am more than happy to change my rating/recommendation.

---

### Official Review · Reviewer_voBL · 2022-07-29

**Originality:** Good
**Technical Quality:** Fair
**Clarity Of Presentation:** Excellent
**Impact:** 3

**Recommendation:**

Weak Reject: I recommend rejecting the paper, but will not argue for my recommendation if the majority of other reviewers have a different opinion.

**Summary:**

The paper focuses on a practical evaluation of how higher-fidelity simulators impact sim2real transfer. It postulates that, contrary to current convention, simulators with high-fidelity are _actually detrimental_ in bridging the sim2real gap. To that end, the authors carried out a substantial amount of simulated and hardware experiments using a hierarchical control policy, where a high-level visual navigation policy is combined with a low-level dynamic controller to perform a Point Goal navigation task.

To corroborate their hypothesis, the authors perform an ablation study using three types of robots and two simulators and two types of simulations: kinematic, where the visual navigation policy running at 1 Hz sets the target linear and angular speeds and the simulator teleports the robot to the new location; and dynamic, where a low-level controller running at 240 Hz receives the target velocities and translates the signal to proper joint configurations and torque control.

**Issues:**

Please refer to points made in the Strengths and Weaknesses section.

**Quality Of The Limitations Section:**

Limitations are not well addressed

**Reviewer Expertise:**

4: The reviewer is confident but not absolutely certain that the evaluation is correct

**Robotics Focus:**

Sufficient demonstration on hardware

**Strengths And Weaknesses:**

The paper is well written and organised, with clear sections following a logical progression. The ablation study required a combinatorial number of computational experiments to be performed, exhausting the various combinations of robots, simulators and simulation types. Furthermore, the paper also presents real robotic experiments, for 15 different runs.

These results are analyzed and used as evidence to support the claim that lower-fidelity simulators are actually offer better sim2real transfer. Although the main idea of the paper is bold and refreshing, I think several points need to be addressed in order for such claim to be validated.

Firstly, the authors use a common wall-clock period for training the visual navigation policy, but recognize that when using dynamical simulation the number of actual experience steps executed by the policy is much lower, due to the increased computational demand. They categorize this as an advantage for policies trained in simpler simulation and as one of the reasons a higher-fidelity simulator leads to worse sim2real transfer. Although this is a valid point from a practical standpoint, it is not a valid argument against the degree of fidelity of the physics simulation. That is to say, would there be a difference in sim2real performance if one could run a higher-fidelity simulation at the same rate as a lower-fidelity one? As it stands, this goes to show that a longer training experience improves the sim2real transfer. Since increasing the training time for dynamic policies is likely unfeasible, the training time for kinematic policies could be reduced to account for the experience difference.

The second point relates to the low-level controllers used. In the kinematic simulation, there is no need for a simulated low-level controller as the robot is simply teleported to the next state. In practice, the visual policy is providing set points for the built-in low-level controllers. On the other hand, for the dynamic simulation, it is necessary to include a simulated low-level controller. In the context of the current work, it was further necessary to use a different kind of simulated low-level in training and evaluation. The authors make the case that this serves as a proxy for the gap between the kinematic simulation and the built-in controllers response. However, this would not be the case the simulated low-level controller has a _biased_ error, since in that case the visual navigation policy would learn to compensate such bias. Since the kinematic simulator only has simulated noise in the euller integration, one can guarantee that error is unbiased. Furthermore, this makes it harder to untangle how much of the degraded performance during evaluation stems from the visual navigation policy or the low-level controllers.

Additionally, with regards to the claim in section 5 that the results presented indicate that the dynamic policies learned are overfitting to the simulator dynamics. I find that due to the two aforementioned points, it is hard to untangle the sources of error and it is not clear that overfitting is the leading cause of degraded sim2real transfer, especially when dynamic policies do not have enough experience to reach a plateau performance.

Finally, although the current work is extensive, it covers a single task which can be reasonably well abstracted. One would imagine that in certain tasks, e.g. pouring or manipulating liquids, a higher-fidelity physics simulation could be more helpful. With that in mind, it would be important to study other tasks before making a general more general claim.

**Summary Of Recommendation:**

The paper is well written and brings a refreshing and impactful new perspective on sim2real transfer. However, there are important points that should be addressed before the paper is ready for publication, especially since it challenges the status-quo of high fidelity simulation.

---

### Official Review · Reviewer_MhUg · 2022-07-31

**Originality:** Fair
**Technical Quality:** Good
**Clarity Of Presentation:** Very Good
**Impact:** 2

**Recommendation:**

Weak Reject: I recommend rejecting the paper, but will not argue for my recommendation if the majority of other reviewers have a different opinion.

**Summary:**

This paper uses two different simulator abstractions to train a high-level navigation policy steering a legged robot around obstacles. The authors compare a kinematic simulation, where the robot is abstracted to a floating base, which is teleported based on the velocity commands, to a dynamical simulation, where a low-level controller is used to actuate the legs of the robot.
After extensive experimentation, the authors conclude that policies trained in the simplified/kinematic simulation are superior to the ones trained in the dynamic setting. They achieve better performance in simulation and transfer better to a real Spot robot.


**Issues:**

See above.

Minor issues:

“...can save simulation … and developer time spent building and designing these controllers.” Wouldn’t these controllers be needed on the real robot anyway?

“In this work, we show that such end-to-end learning is also possible for complex, legged robots.” -> what is the difference between developing a navigation module for a wheeled or legged platform if the low-level controller is provided?

[40] is not a blind policy.

In section 5, the authors add noise to the translation of the kinematic policy, thus showing that they need to close the sim2real to improve performance. This contradicts the message of the paper.


**Quality Of The Limitations Section:**

Limitations are not well addressed

**Reviewer Expertise:**

4: The reviewer is confident but not absolutely certain that the evaluation is correct

**Robotics Focus:**

Sufficient demonstration on hardware

**Strengths And Weaknesses:**

The authors conduct extensive experiments with a clearly presented and technically sound approach. However, I think the results are not as novel and surprising as presented. The real conclusion of the paper is that kinematic simulation is enough to train a navigation policy. This is not surprising since a legged robot with a good locomotion controller is an omnidirectional platform that can be steered with velocity commands. This abstraction has been shown to transfer easily to the real world before [1] (this work should have been cited)
Two simple reasons can explain the inferiority of the dynamically trained policies.
1) As the authors explain, the dynamical simulation is much slower, so the policies used in the comparison had access to much less training data. In that case, why would we expect them to perform similarly?
2) The authors use simplistic low-level controllers which “are unstable at higher velocities”. The left plot of figure 6 shows that kinematic policies also fail when deployed in the dynamical simulation, while they work much better in the real world. This suggests that the low-level controllers used in simulation are inferior to the real one provided by BD. If the robots keep falling in simulation, the high-level policy will struggle to learn.

[1]: Hoeller, D., Wellhausen, L., Farshidian, F., & Hutter, M. (2021). Learning a State Representation and Navigation in Cluttered and Dynamic Environments. IEEE Robotics and Automation Letters, 6(3), 5081–5088. https://doi.org/10.1109/LRA.2021.3068639

**Summary Of Recommendation:**

While the experiments are relevant to the robotics community. I believe that the conclusions drawn by the authors are not well founded and should not be published in the current state.

---

### Official Review · Reviewer_Soyb · 2022-08-01

**Originality:** Fair
**Technical Quality:** Good
**Clarity Of Presentation:** Good
**Impact:** 3

**Recommendation:**

Weak Accept: I recommend accepting the paper, but will not argue for my recommendation if the majority of other reviewers have a different opinion.

**Summary:**

The authors hypothesize that a decrease in robot simulation fidelity may result in a higher efficacy sim2real transfer of deep reinforcement learning policies for visual navigation tasks. A dichotomy is drawn between a low-frequency kinematic simulation environment (low-fidelity simulation) and a high-frequency dynamic simulation environment (high-fidelity).

This hypothesis is tested via training on 3 simulated quadrupedal robots across 2 simulation environments, and testing on one vehicle in the real-world setting.

Robot behavior is governed by a hierarchical control architecture, with a high-level policy mapping depth imagery and GPS+compass measurements to velocity commands, and one of two low-level controllers (kinematic and dynamic) mapping velocity to a center-of-mass goal position or direct joint angles.

The authors find that use of kinematic-only simulation environments result in stronger sim2real transfer given equal training time.

**Issues:**

Given the nature of this conference's 8-page paper limit, much of the below would be well-suited to inclusion in the paper's abstract, should space constraints prove problematic.

1. Additional discussion of kinematic-dynamic environment tradeoff: The work would be improved by addressing conditions under which kinematic simulation is unsuitable for sim2real transfer.  For the PointGoal Navigation task focused upon in this work, it is clear that simulation at kinematic fidelity is sufficient to produce an efficacious policy for real-world operation. However, there are many other task conditions in which robot kinematics alone are insufficient for commanding a vehicle in real-world environments. These should be noted and discussed.

2. Description of high-level policy parameters: The work is very limited in its description of the exact architecture and parameterization of the high-level policy and reward function. Additional mathematical description would be beneficial.

3. Clarify additions to work in Ref. [22] (Truong et al.): Much of the architecture and experimental setup appears to bear strong similarities to work put forward in reference 22 of the paper. A more explicit disambiguation of changes made in this work over Ref. 22 would serve to make clear why some of the concerns regarding dynamic modeling in that work do not apply here. Furthermore, some implementation details may bear repeating in this paper, as elements present in that work (such as mathematical rigor) are cited, but not expanded upon in a thorough way.

**Quality Of The Limitations Section:**

Limitations are not well addressed

**Reviewer Expertise:**

4: The reviewer is confident but not absolutely certain that the evaluation is correct

**Robotics Focus:**

Sufficient demonstration on hardware

**Strengths And Weaknesses:**

- Strengths

The work applies a thorough experimental methodology, examining performance of learned policies in simulation and on a real-world Spot quadrupedal robot.  Performance comparison across multiple simulators (Habitat and iGibson) provides an additional degree of thoroughness, providing insight into current state-of-the-art simulation tools, as well as preventing limitations in a single simulator implementation from biasing the paper's results.

- Weaknesses
  - Insufficient discussion of negative effects of kinematic
    simulation:

The need for agent knowledge of vehicle dynamics is repeatedly cited in the sim2real robotics literature, especially for complex motion platforms such as quadrupedal robots.  This is due to the lack of conformity of idealized vehicle kinematics to a real-world dynamic multi-legged system.  The paper in its current form fails to thoroughly address why the proposed system is immune to this constraint, and neglects to discuss task conditions under which a policy learned in kinematic simulation would be inapplicable.

  - Artificial penalization of higher-complexity simulation:

A key claim in this work is the stark contrast in efficacy and generalization of policies trained in physics-absent kinematic environments versus those in high-fidelity dynamic environments. However, this contrast is primarily due (as noted in Section 5) to the "poverty" of experience provided to agents trained in a dynamic environment.  The quantity of training provided to agents is controlled by wall clock time of the training task, resulting in 1/20th of the experience steps available to dynamic agents compared to their kinematic equivalent.  Benchmarking performance as a function of training wall-clock weakens the claim that lower fidelity simulation yields stronger sim2real transfer, instead reinforcing the need for sufficient experience for robust learned policy development.  Should the comparison been drawn between high and low fidelity training environments under the condition of equal steps of experience, stronger claims could be made regarding  the benefit of low-fidelity training environments.

However, the contrast in requisite training time between low and high fidelity simulation is still important to note.  One could forsee a compelling claim being made for scrutinizing the costs and benefits of higher fidelity simulation, drawing a conclusion that under some task conditions, high-fidelity learning environments are needlessly expensive computationally.  Nonetheless, the wide claim of the higher sim2real transfer for lower fidelity simulation is not fully supported in the work as it stands.

**Summary Of Recommendation:**

On the whole, this work provides a reasonable case for the use of lower-fidelity simulation for use in sim2real transfer. However, there are some crucial points that are analyzed in such a way as to limit the strength of the claims made. Despite relatively strong technical quality and thoroughness experimentally, there remain a select few outstanding questions on the part of the reviewer which provoke hesitation in recommending acceptance.

Post-Rebuttal: The authors' revisions have strengthened the paper and addressed several of the original concerns expressed above, and the rating for this paper has been amended accordingly.

---

### Meta-Review · Area_Chair_feAk · 2022-08-12

**Recommendation:** Accept (Poster)
**Confidence:** 5

**Metareview:**

The reviewers commented positively about the general hypothesis discussed in the paper (lower simulation fidelity can lead to higher sim2real success) and the number of experiments reported which includes multiple simulators and platforms. The main issues to be address by the authors in the rebuttal phase are:

1. Discuss negative effects of kinematic simulation
2. Discuss the artificial penalization of higher-complexity simulation
3. Discuss the relationship with the work in "Hoeller, D., Wellhausen, L., Farshidian, F., & Hutter, M. (2021). Learning a State Representation and Navigation in Cluttered and Dynamic Environments. IEEE Robotics and Automation Letters" and in ref [22]
4. Discuss the low level controller used and the implications to policy learning
5. Discuss the conclusion that policies trained with dynamic simulation "overfit" to the simulator dynamics in policies that have not converged yet

=========================================
Post rebuttal update

The authors have strengthened the paper and addressed most of the concerns from the reviewers. I believe this is a thought provoking paper that will generate interesting discussions at the conference and should therefore be accepted.

**Best Paper Nomination:**

No